# Calaxin is a key factor for calcium-dependent waveform control in zebrafish sperm

Motohiro Morikawa ⓘ, Hiroshi Yamaguchi ⓘ, Masahide Kikkawa ⓘ

**Calcium is critical for regulating the waveform of motile cilia and flagella. Calaxin is currently the only known molecule involved in the calcium-dependent regulation in ascidians. We have recently shown that Calaxin stabilizes outer arm dynein (OAD), and the knockout of Calaxin results in primary ciliary dyskinesia phenotypes in vertebrates. However, from the knockout experiments, it was not clear which functions depend on calcium and how Calaxin regulates the waveform. To address this question, here, we generated transgenic zebrafish expressing a mutant E130A-Calaxin deficient in calcium binding. E130A-Calaxin restored the OAD reduction of $calaxin^{-/-}$ sperm and the abnormal movement of $calaxin^{-/-}$ left–right organizer cilia, showing that Calaxin's stabilization of OADs is calcium-independent. In contrast, our quantitative analysis of E130A-Calaxin sperms showed that the calcium-induced asymmetric beating was not restored, linking Calaxin's calcium-binding ability with an asymmetric flagellar beating for the first time. Our data show that Calaxin is a calcium-dependent regulator of the ciliary beating and a calcium-independent OAD stabilizer.**

## Introduction

Motile cilia/flagella are antenna-like organelles protruding from the cell surface and have been conserved among eukaryotes (Satir & Christensen, 2007). They show various types of beating optimized for their functions, such as the generation of extracellular fluid flow and the locomotion of spermatozoa. Vertebrate left–right organizer (LRO) has conically rotating cilia to generate leftward fluid flow, leading to left–right patterning during embryogenesis (Essner et al, 2002, 2005). Sperm flagella show planar beatings for their locomotion (Lindemann & Lesich, 2021). In many species, they change the asymmetry of their beating to control the swimming path and reach eggs directed by chemical cues (Yoshida & Yoshida, 2011). The cytoskeletal structure of cilia is called axoneme, which has nine peripheral doublet microtubules (DMTs) with or without two central microtubules, where axonemal dyneins generate the beating force (Ishikawa, 2017).

Intraciliary calcium plays a key role in regulating the waveform of sperm flagella. High (0.001–0.1 mM) calcium condition induces asymmetric waveform in demembranated sperm of sea urchin (Brokaw et al, 1974; Brokaw, 1979), ascidian *Ciona intestinalis* (Mizuno et al, 2009), and rats (Lindemann & Goltz, 1988). Photolysis of caged calcium in zebrafish sperm results in highly asymmetric beating waveforms and curved swimming paths (Fechner et al, 2015). Fluctuation of intraciliary calcium concentration coincides with an asymmetric beating in intact sea urchin sperm (Shiba et al, 2008; Alvarez et al, 2012). In mammals, sperm-specific calcium channel CatSper and calcium influx are necessary for hyperactivation, which involves the asymmetric waveform and a higher beating frequency (Chang & Suarez, 2010).

These studies indicate calcium is essential for sperm to control their waveform, and calcium-dependent dynein regulators have been explored. One such protein is Calaxin, a dynein-associated neuronal calcium sensor family protein conserved among opisthokonts (Mizuno et al, 2009). Calaxin was isolated as a component of outer arm dynein (OAD) of *C. intestinalis* sperm, and recent studies of cryo-electron microscopy have shown that Calaxin is a component of the OAD docking complex (DC), a linker structure that tethers OAD to the peripheral DMT (Gui et al, 2021; Yamaguchi et al, 2023). Our group previously generated and analyzed Calaxin knockout mice and zebrafish to show that loss of Calaxin caused abnormal movement of sperm flagella and tracheal cilia and defects of LRO ciliary formation (Sasaki et al, 2019). In addition, $calaxin^{-/-}$ zebrafish sperm partly lacked OAD (Yamaguchi et al, 2023). These results suggest that Calaxin is an indispensable component of vertebrate motile cilia that stabilizes the binding of OAD to DMTs. Recently, its mutation was also discovered in primary ciliary dyskinesia (PCD) patients exhibiting laterality defects and respiratory symptoms (Hjeij et al, 2023).

The relationship between Calaxin's calcium-binding ability and its functions, including the flagellar waveform regulation, has not been fully investigated. In invertebrates, Calaxin inhibits axonemal dynein in a calcium-dependent manner. Repaglinide, a neuronal calcium sensor family protein inhibitor, and anti-Calaxin antibody inhibit chemotactic movements and the calcium-induced asymmetric waveform in *C. intestinalis* sperm. Calaxin also inhibits OAD activity in the presence of calcium in vitro, and calcium-binding–

Department of Cell Biology and Anatomy, Graduate School of Medicine, The University of Tokyo, Tokyo, Japan

Correspondence: mkikkawa@m.u-tokyo.ac.jp

deficient Calaxin mutant shows incomplete inhibition (Mizuno et al, 2012; Shiba et al, 2023). However, it is undetermined whether the calcium-binding ability of Calaxin contributes to the calcium-dependent asymmetric ciliary beating and the OAD-stabilizing function. In addition, it is also unexplored whether its calcium-binding ability is conserved in vertebrates.

To analyze the calcium-dependent and calcium-independent functions of Calaxin, we observed $calaxin^{-/-}$ zebrafish and zebrafish expressing WT-Calaxin or calcium-binding–deficient E130A-Calaxin. As we showed recently (Yamaguchi et al, 2023), $calaxin^{-/-}$ zebrafish sperm flagella contained less amount of OAD, indicating that Calaxin is one of the DC components that stabilizes OAD. WT- and E130A-Calaxin stabilized OAD in sperm flagella, and their expression rescued the abnormal motility of $calaxin^{-/-}$ LRO cilia. However, sperm flagella containing E130A-Calaxin failed to exhibit the asymmetric waveform in response to calcium, whereas those with WT-Calaxin increased asymmetry. Our data succeeded in describing the calcium-dependent and calcium-independent functions of Calaxin in the vertebrate axoneme and directly linked the calcium-binding ability of Calaxin to the calcium-induced asymmetric waveform of sperm flagella.

## Results and Discussion

### *calaxin* mutation causes OAD partial loss in zebrafish axoneme

Previously, we reported that $calaxin^{-/-}$ zebrafish exhibit abnormal KV ciliary movement and consequent randomized left–right patterning (Sasaki et al, 2019). Calaxin is known as a component of the vertebrate DC, the linker structure between OAD and DMT. Thus, we speculate that the $calaxin^{-/-}$ phenotype is caused by OAD defects.

To clarify this point, we investigated OAD localization in $calaxin^{-/-}$ sperm flagella. Immunofluorescence microscopy (IFM) showed that $calaxin^{-/-}$ sperm lacked Calaxin and OAD heavy chains Dnah8 and Dnah9 (Fig 1A, C, and D). Inner arm dynein (IAD) heavy chain Dnah2 showed normal localization (Fig 1B). Interestingly, the borders of OAD(+) and OAD(−) regions varied between $calaxin^{-/-}$ sperms (Fig 1C and D; white lines and white dotted lines show OAD(+) and OAD(−) regions, respectively). These data suggest that *calaxin* mutation caused unstable OAD-DMT docking in sperm flagella and possibly in KV cilia. Consistent with our IFM data, a recent study of PCD patients also showed that *CALAXIN* mutation resulted in the absence of OAD from the distal axoneme (Hjeij et al, 2023). Moreover, our structural analysis of the $calaxin^{-/-}$ axoneme showed that Calaxin is required to stabilize the OAD docking onto DMT in vertebrates (Yamaguchi et al, 2023).

### Zebrafish Calaxin binds to calcium, which is disrupted by the E130A mutation

Calaxin is also known as an OAD-associated calcium sensor. In ascidian *C. intestinalis*, Calaxin plays an essential role in sperm waveform regulation by inhibiting axonemal dynein activity depending on calcium concentration (Mizuno et al, 2012). However, in vertebrates, the calcium-binding ability of Calaxin and its function have not been assessed.

To analyze this point, we focused on zebrafish E130A-Calaxin, which corresponds to *Ciona* E118A-Calaxin. Calaxin has three calcium-binding domains, EF-hands (Fig 1E). Although all three EF-hands of *Ciona* Calaxin bind to a calcium ion, E118A, a single substitution within the EF-hand2 leads to the complete loss of calcium-binding ability (Shojima et al, 2018).

To validate the calcium-binding ability of Calaxin in vertebrates, we purified zebrafish WT- and E130A-Calaxin and performed the isothermal titration calorimetry (ITC) experiment. The result of WT-Calaxin was fitted with a three-site sequential binding model, indicating that all three EF-hands bind to calcium. On the contrary, E130A-Calaxin showed no significant binding, as in *Ciona* E118A-Calaxin (Fig 1F). SDS–PAGE under the presence of calcium or EGTA also revealed that calcium increased the mobility of WT-Calaxin, but E130A-Calaxin showed only a slight change (Fig 1G). This behavior is identical to Calmodulin and its calcium-binding–deficient mutant, respectively (Geiser et al, 1991). To confirm that E130A-Calaxin retains the normal secondary structure, we assessed the interaction between recombinant Calaxin and $calaxin^{-/-}$ axoneme. Recombinant WT- and E130A-Calaxin specifically bound to the OAD(+) region of $calaxin^{-/-}$ sperm flagella, demonstrating that they folded properly (Fig S1A–C). $armc4^{-/-}$ axoneme was used as the negative control because it completely lacks OAD (Yamaguchi et al, 2023). These results indicate zebrafish Calaxin binds to three calcium ions, and E130A-Calaxin completely loses its calcium-binding ability without disrupting the native conformation. EF-hand2 containing zebrafish E130 residue is highly conserved among *Ciona*, zebrafish, and mammals, suggesting the importance of this domain (Fig 1E). Here, we decided to use E130A-Calaxin to analyze the calcium-dependent function of Calaxin.

### Calaxin stabilizes OAD independent of its calcium-binding ability

We previously showed the loss of Calaxin caused the reduction of OAD in sperm flagella and the abnormal Kupffer's vesicle (KV) ciliary movement, resulting in laterality randomization (Sasaki et al, 2019; Yamaguchi et al, 2023). To analyze whether the calcium-binding ability of Calaxin is involved in these phenotypes, we performed rescue experiments using WT- and E130A-Calaxin.

First, we assessed the sperm phenotype by generating transgenic zebrafish expressing WT- or E130A-Calaxin under beta-actin promoter (*Tg(actb2:calaxin);calaxin^{-/-}* and *Tg(actb2:calaxin_E130A); calaxin^{-/-}*). Hereafter, we call them *Tg WT* and *Tg E130A*, respectively. Transgenes were integrated into the $calaxin^{-/-}$ zebrafish with the Tol2 transposon system (Kawakami, 2007), resulting in viable and fertile animals. IFM showed that both WT- and E130A-Calaxin were properly incorporated into the axoneme of transgenic sperm (Fig 1A). Despite the loss of calcium-binding ability, E130A-Calaxin, as well as WT-Calaxin, restored the loss of OAD in the distal region of $calaxin^{-/-}$ sperm flagella (Fig 1C and D). These results show that Calaxin localizes to the axoneme and stabilizes OAD independent of its calcium-binding ability. Note that OAD was absent from all distal tips of sperm flagella obtained from both transgenic animals (Fig 1C and D, white boxes and magnified views). We speculate that this is caused by different expression regulations between the native *calaxin* promoter and the *actb2* promoter we used for transgene expression.

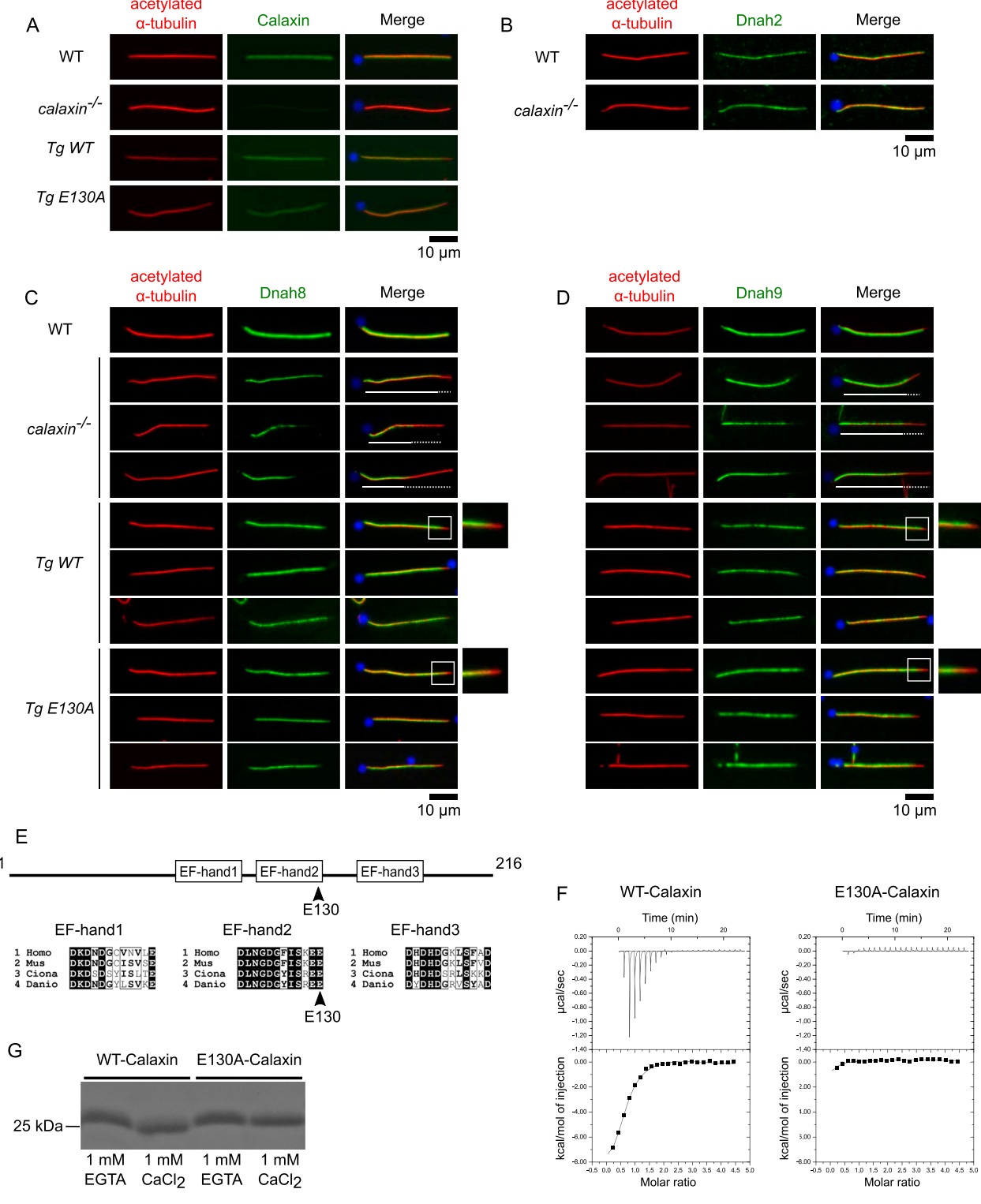

**Figure 1. Calaxin calcium-binding activity is dispensable to stabilize outer arm dynein (OAD).**
**(A, B, C, D)** Immunofluorescence of WT, *calaxin⁻/⁻*, *Tg(actb2:calaxin);calaxin⁻/⁻*(*Tg WT*), and *Tg(actb2:calaxin_E130A);calaxin⁻/⁻*(*Tg E130A*) sperm. Sperm were stained with anti-Calaxin (A), Dnah2 (B), Dnah8 (C), or Dnah9 (D) antibodies (green) and costained with DAPI (blue) and acetylated α-tubulin (red). Scale bars, 10 μm. **(A)** *calaxin⁻/⁻* sperm flagella did not contain Calaxin, whereas *Tg WT* and *Tg E130A* restored this phenotype. **(B)** *calaxin⁻/⁻* sperm flagella contained Dnah2 as in WT. **(C, D)** *calaxin⁻/⁻* sperm flagella lacked Dnah8 and Dnah9 on their distal half, whereas WT sperm contained them along the whole length of the axoneme. The amount of Dnah8 and Dnah9 was variable between *calaxin⁻/⁻* flagella. White lines and dotted lines show OAD(+) and OAD(−) regions, respectively. *Tg WT* and *Tg E130A* contained Dnah8 and Dnah9 along the length of the axoneme except for distal tips (white boxes), almost restoring the phenotype of the *calaxin⁻/⁻* mutant. The magnified images of distal tips (white boxes) are

Next, we also tested the KV ciliary phenotype. *calaxin*$^{-/-}$ KV cilia exhibited a decreased ratio of regularly rotating cilia and a lower beating frequency. Injection of either *calaxin* WT or E130A mRNA into *calaxin* $^{-/-}$ embryos restored the motility of KV cilia. Importantly, the fraction of the regularly rotating cilia and the beating frequency were identical between WT and E130A mRNA injections, indicating that E130A mutation does not alter KV ciliary motility (Fig 2A–C and Video 1). In addition, *Tg E130A* zebrafish showed normal laterality (Fig 2D). These results indicate that Calaxin contributes to KV ciliary motility independent of its calcium-binding ability, possibly stabilizing OAD.

### Demembranated sperm model reconstituted Calaxin-dependent calcium-induced waveform asymmetry

The inhibition of Calaxin disrupts sperm's calcium-induced asymmetric beating (Mizuno et al, 2012), but it is undetermined whether Calaxin directly detects calcium or is a mediator of another calcium sensor. To test whether its calcium-binding ability is essential for an asymmetric beating, we established a demembranated sperm model according to the *Chlamydomonas* flagellum model (Brokaw, 1979; Kamiya & Witman, 1984). Briefly, sperm was demembranated with 0.5% NP-40, then reactivated by 50 $\mu$M ATP + 50 $\mu$M ADP in the presence of 0.1 mM EGTA (EGTA condition) or CaCl$_2$ (pCa4 condition). ADP was supplemented to activate all axonemal dynein subspecies (Yagi, 2000). A comparison of different ATP and ADP concentrations showed that a higher ATP concentration induced a faster beating frequency and that the supplementation of ADP reduced the variation of beating frequencies (Fig S2A). On the contrary, a higher ATP concentration resulted in a lower reactivation rate, especially in the pCa4 condition (Fig S2B). Thus, we chose the 50 $\mu$M + ATP 50 $\mu$M ADP condition to enable the stable beating frequency and the higher reactivation rate. WT, *calaxin* $^{-/-}$, *Tg WT*, and *Tg E130A* sperm models were successfully reactivated.

In WT, *Tg WT*, and *Tg E130A* sperm models, the pCa4 condition induced more asymmetric waveform than the EGTA condition, but the asymmetry of *Tg E130A* was lower (Fig 3A–C and Video 2). *Calaxin* $^{-/-}$ sperm model contained a proximal region with high amplitude and a distal region with low amplitude (Fig 3D and Video 2). We investigate this phenotype in the later section. Beating frequencies of *Tg WT* and *Tg E130A* sperm models were similar in EGTA and pCa4 conditions, respectively. Sperm models showed a slower beating under the pCa4 condition, although it was not statistically significant in *calaxin* $^{-/-}$ and *Tg WT* (Fig S2C). To test whether Calaxin's calcium-binding ability is necessary for calcium-induced waveform transformation, we decided to quantify the waveform asymmetry of WT, *Tg WT*, and *Tg E130A* sperm models.

### Calaxin E130A mutant diminished calcium-induced waveform asymmetry

The planar ciliary beating can be divided into the static component and the dynamic component corresponding to the time-averaged basal shape and the dynamic beating, respectively (Eshel & Brokaw, 1986; Geyer et al, 2016). To quantify the sperm waveform asymmetry, we calculated the spatial average of the static component in each flagellum, which we call "basal curvature" below (Fig S2E). We traced sperm flagella and calculated their tangent angles. Because zebrafish sperm have round heads, we could not calculate the tangent angle relative to the direction of the sperm head as in sea urchin sperm (Brokaw, 1979). Instead, we selected sperm with their heads fixed to the glass slide and freely swimming flagella and calculated the tangent angle relative to the recorded frame. To obtain the time and spatial average, we plotted tangent angles from $\geqq$ 5 traces over one beat cycle followed by the fitting with the linear equation. Basal curvature was defined as the slope of the fitted equation.

The quantification revealed that the *Tg E130A* sperm model failed to increase the basal curvature with statistical significance in response to calcium, whereas WT and *Tg WT* significantly elevated it in the pCa4 condition. The basal curvature of *Tg E130A* in the pCa4 condition was also significantly lower than *Tg WT* in the same condition (Fig 3E). To test whether beatings are successfully separated into dynamic and static components, we calculated the dynamic component by subtracting the basal curvature from the tangent angles. All of WT, *Tg WT*, and *Tg E130A* showed uniform basal curvature and symmetric, sinusoidal dynamic components, which confirmed that their beatings were successfully separated into the dynamic and static components (Fig 3A–C, dotted lines, Fig S2F). Because the calculated dynamic components of *Tg WT* and *Tg E130A* were symmetric, the loss of OADs at the distal tips of their flagella did not strongly affect the basal curvature. We also calculated the asymmetry index as in *Ciona* sperm (Mizuno et al, 2012) (Fig S2D). The result showed a similar trend as the basal curvature: the pCa4 condition led to higher asymmetry indices, with *Tg E130A* exhibiting a weaker elevation compared with WT and *Tg WT*. However, these differences did not reach statistical significance, possibly because of the limited sample size. Note that we used an alternative definition of "P-bend" and "R-bend": we defined the beating with higher curvature as "P-bend" and the other as "R-bend," deviating from the original definition based on the sperm swimming path, as we observed sperm fixed to the glass slide. These results demonstrate that Calaxin's calcium-binding ability is necessary to generate the calcium-induced asymmetric waveform.

---

also shown on the right. **(E)** Domain structure of Calaxin (top) and multiple alignments of three EF-hand calcium-binding regions (bottom). E130 (arrowheads) is the last residue of the EF-hand2, which is the most highly conserved between *Homo sapiens*, *Mus musculus*, *Ciona intestinalis*, and *Danio rerio*. **(F)** Isothermal titration calorimetry of recombinant WT-Calaxin (left) and E130A-Calaxin (right). Three sequential binding site models were used for fitting. WT bound to three calcium ions per molecule, whereas E130A showed no binding. **(G)** SDS–PAGE of recombinant WT-Calaxin and E130A-Calaxin with 1 mM EGTA or CaCl$_2$. WT-Calaxin showed higher mobility in 1 mM CaCl$_2$ compared with 1 mM EGTA, whereas E130A-Calaxin showed a slight difference.

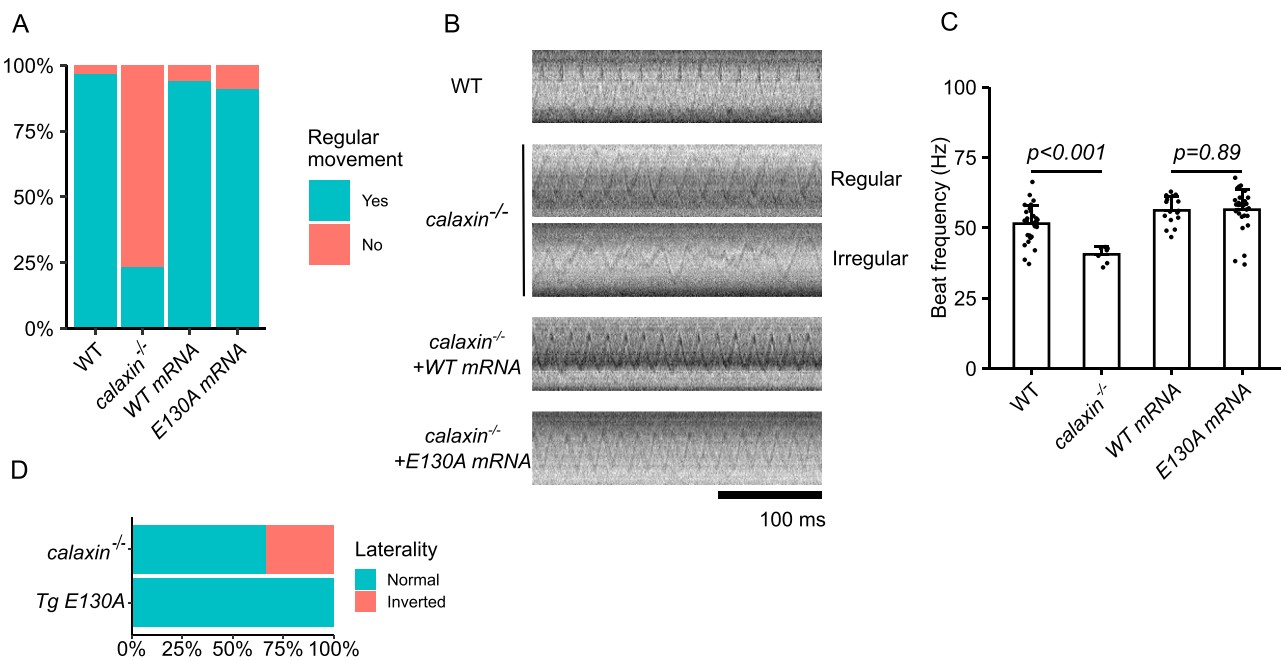

**Figure 2. Calaxin calcium-binding activity is dispensable for KV ciliary movement.**
**(A)** Classification of KV cilia. KV cilia of WT embryos, *calaxin*⁻/⁻ embryos, and *calaxin*⁻/⁻ embryos injected with *calaxin* WT (*WT mRNA*) or E130A mRNA (*E130A mRNA*) were recorded by a bright-field microscope (AF6000B; Leica) equipped with a high-speed camera (HAS-L1; DITECT) at 1,000 fps at the 8–10 somite stage. KV ciliary motility was classified into normal conically rotating movement and abnormal movement, including irregular movement and quiescence. Both WT and E130A mRNA restored the irregular movement of *calaxin*⁻/⁻ KV cilia. All data were collected from n = 30 cilia from 6 embryos (WT), 30 cilia from 5 embryos (*calaxin*⁻/⁻), 17 cilia from 3 embryos (*WT mRNA*), and 33 cilia from 4 embryos (*E130A mRNA*). **(B)** Kymographs of KV cilia. Normal regularly rotating cilia from WT, *calaxin*⁻/⁻, and *calaxin*⁻/⁻ injected with WT and E130A mRNA embryos and an irregularly moving cilium from *calaxin*⁻/⁻ embryo are shown. **(C)** Beat frequencies of KV cilia. Normal conically rotating cilia shown in (A) were subjected to the analysis. *calaxin*⁻/⁻ cilia showed a slower beating frequency. Both calaxin WT and E130A mRNA injected into *calaxin*⁻/⁻ embryo restored the motility to the same extent. All data are shown with a mean (bar graphs) + SE (error bars). Scale bar, 100 ms. n = 29 cilia from 6 embryos (WT), seven cilia from 3 embryos (*calaxin*⁻/⁻), 16 cilia from 3 embryos (*calaxin*⁻/⁻ + WT mRNA), and 30 cilia from 4 embryos (*calaxin*⁻/⁻ + E130A mRNA). **(D)** Laterality of the *calaxin*⁻/⁻ and *Tg E130A* embryos. *calaxin*⁻/⁻ and *Tg E130A* adults were crossed to obtain embryos. The direction of the heart looping was observed at the 30–33 hpf stage to determine the laterality, followed by genotyping. *calaxin*⁻/⁻ embryos exhibited laterality randomization as observed before, whereas *Tg E130A* showed normal laterality. n = 12 embryos for each genotype.

### *calaxin*⁻/⁻ sperm model has a highly asymmetric and inactive distal region

*calaxin*⁻/⁻ sperm model contained an active proximal region and an inactive distal region. To further investigate this phenotype, we compared WT and *calaxin*⁻/⁻ sperm models. The distal region of the *calaxin*⁻/⁻ sperm model showed higher asymmetry than that of WT in the pCa4 condition, but both genotypes showed similar waveforms in the proximal region (Fig 3A and D and Video 2). Moreover, the dynamic component of the *calaxin*⁻/⁻ sperm model was not uniform, indicating that the proximal and distal regions show different beating patterns (Fig S2F). This distal region seemed to coincide with the distal OAD(–) IAD(+) region shown in the IFM experiment. The length of the inactive region differed between flagella, possibly reflecting the variable length of the OAD(–) region (Fig 1C and D). From these results, we speculated that IAD affects waveform asymmetry calcium-dependently.

### IAD contributes to calcium-dependent flagellar asymmetry

To further test the contribution of IAD to the asymmetric waveform, we observed the *armc4*⁻/⁻ zebrafish sperm model and quantified the basal curvature. Armc4 is a major component of the OAD-DC

(Gui et al, 2021), and *armc4*⁻/⁻ zebrafish sperm axoneme does not contain OAD, whereas it retains IAD and other DMT appendages (Yamaguchi et al, 2023). *Armc4*⁻/⁻ sperm model showed highly curved flagella in the EGTA condition, and the pCa4 condition further increased the curvature (Fig S3A and B and Video 3). Most of the *armc4*⁻/⁻ sperm model was immotile or only showed irregular, vibrating movement (Fig S3C and Video 3). The addition of orthovanadate, a dynein ATPase inhibitor, resulted in straight and immotile flagella in both EGTA and pCa4 conditions, suggesting that ATP-dependent mechanisms generate such curvatures (Fig S3A and B and Video 3). To quantify the basal curvature of immotile or irregularly beating flagella, we randomly selected one frame from each video and traced the flagellum. These results reinforce the idea that IAD alters waveform asymmetry calcium-dependently.

### Calaxin works as a calcium-dependent dynein regulator and a calcium-independent dynein stabilizer

Our results showed that zebrafish Calaxin has three calcium-binding sites that are necessary for the calcium-induced asymmetric beating but dispensable to stabilize OAD. Only the sperm model expressing calcium-binding–deficient E130A-Calaxin showed

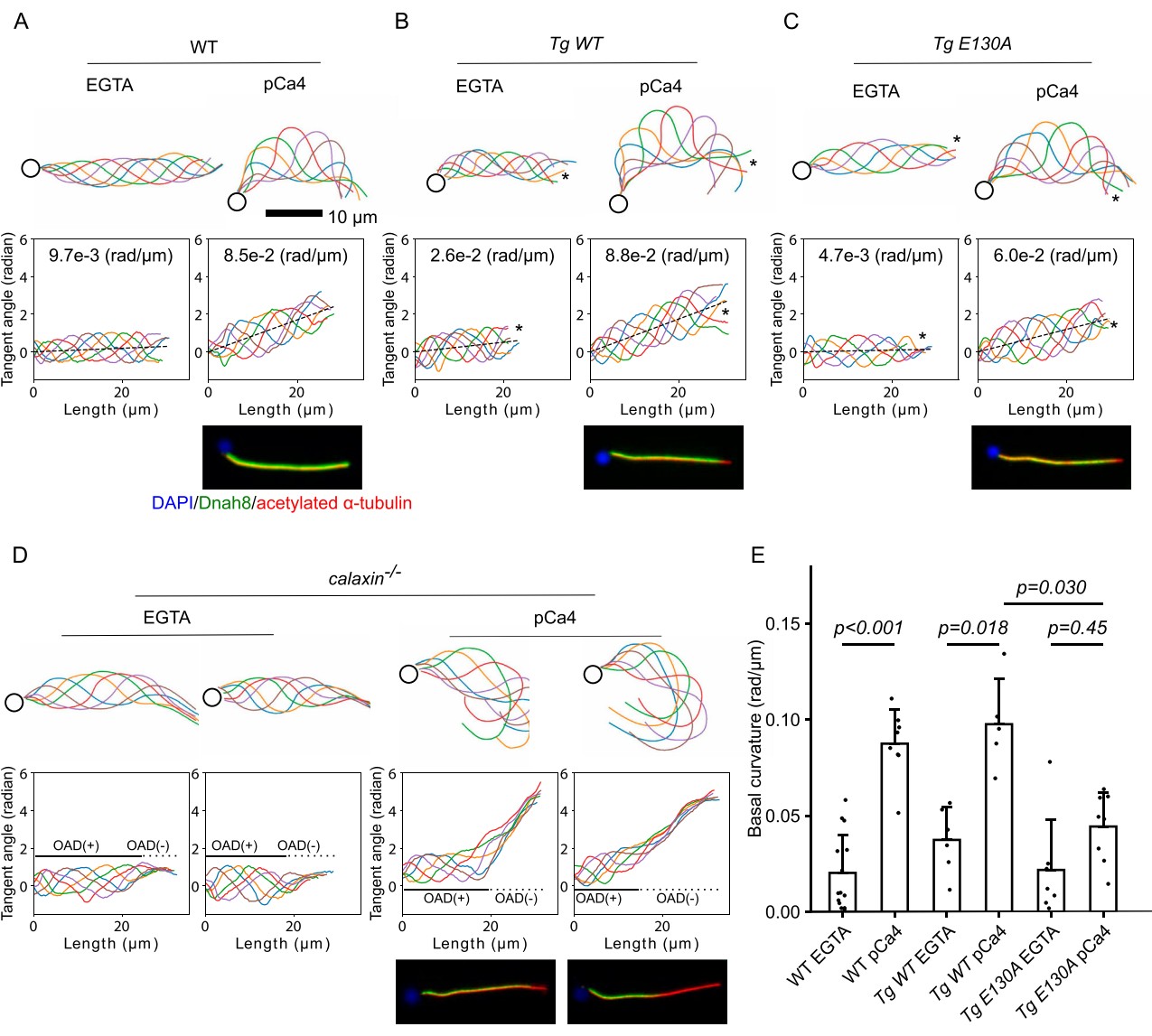

**Figure 3. Calaxin calcium-binding activity is necessary for the calcium-induced asymmetric beating of sperm.**
**(A, B, C, D)** Traces (top) and tangent angle plots (bottom) of WT (A), *Tg WT* (B), *Tg E130A* (C), and *calaxin*$^{-/-}$ (D) demembranated sperm models under the presence of 0.1 mM EGTA (EGTA) or CaCl$_2$ (pCa4). Immunofluorescence microscopy results from Fig 1C are also shown for the comparison. WT and *Tg WT* showed an almost symmetric waveform in the EGTA condition, and the pCa4 condition induced asymmetric waveform. *Tg E130A* showed a similar waveform in the EGTA condition, but the pCa4 condition induced weaker asymmetry compared with WT and *Tg WT*. *calaxin*$^{-/-}$ sperm model had an active proximal region and an inactive distal region. The inactive distal region exhibited a highly curved waveform in the pCa4 condition. **(A, B, C)** Calculated basal curvatures are shown in tangent angle plots. **(B, C)** Asterisks indicate distal ends of flagella that lack an outer arm dynein (OAD). **(D)** Black lines and dotted lines show estimated OAD(+) and OAD(−) regions, respectively. **(E)** Comparison of basal curvatures of WT, *Tg WT*, and *Tg E130A* sperm flagella. The pCa4 condition induced significantly higher basal curvature in both WT and *Tg WT*, but *Tg E130A* did not show a significant elevation of basal curvature. In addition, the basal curvature of *Tg E130A* was significantly lower than that of *Tg WT* in the pCa4 condition. All data are shown with a mean (bar graphs) + SE (error bars). All data were collected from n = 14 (WT EGTA), 8 (WT, pCa4), 6 (*Tg WT* EGTA), 5 (*Tg WT* pCa4), 7 (*Tg E130A* EGTA), and 8 (*Tg E130A* pCa4) sperm flagella from two or more independent experiments.

an incompletely asymmetric waveform in response to calcium, whereas those expressing WT-Calaxin exhibited a normal asymmetric beating. Combining this with the ITC experiment, our results directly linked Calaxin's calcium-binding ability to the calcium-induced asymmetric beating. Considering that *Ciona* Calaxin inhibits OAD activity in response to calcium (Mizuno et al, 2012), it appears Calaxin binds to calcium and inhibits OAD to elevate the waveform asymmetry when intraciliary calcium concentration increases.

We found that the *Tg E130A* sperm model exhibited a uniform and symmetric dynamic component even in the presence of calcium, suggesting that Calaxin's calcium-binding ability does not influence the dynamic beating (Fig S2F). This is inconsistent with the previously proposed model, where Calaxin influences the bend formation and propagation and triggers an asymmetric beating calcium-dependently (Mizuno et al, 2012; Shiba et al, 2023). Instead, our results imply that Calaxin's calcium-binding ability is necessary to regulate the basal curvature but dispensable for the bend

propagation. The difference may result from different model species, zebrafish we used and *C. intestinalis* in existing studies. Alternatively, the E130A-Calaxin we used might more selectively disrupt the calcium-dependent functions of Calaxin while maintaining its calcium-independent functions.

Our results also suggested that Calaxin is not the sole calcium-dependent axonemal dynein regulator. Both WT- and E130A-Calaxin–expressing sperm models showed a slower beating under the presence of calcium (Fig S2C). Moreover, calcium-induced weak asymmetry was observed in the E130A-Calaxin–expressing sperm model, although it was not statistically significant (Fig 3C and E). These results imply that there are calcium-dependent regulators other than Calaxin. IAD is one of the candidates because calcium elevated the asymmetry of the OAD(–) *armc4*$^{-/-}$ sperm model and the distal region of the *calaxin* $^{-/-}$ sperm model (Figs 3D and S3A and B). Axoneme also contains other calcium-binding proteins, such as calmodulin (Pazour et al, 2005).

Another important finding is that Calaxin works as a calcium-independent OAD stabilizer and a calcium-dependent dynein regulator. Sperm flagella derived from *calaxin* $^{-/-}$ zebrafish partly lack OAD heavy chains Dnah8 and Dnah9. Intriguingly, we have previously shown that OADs are also reduced in the proximal region of *calaxin*$^{-/-}$ flagella (Yamaguchi et al, 2023). This indicates that Calaxin knockout destabilizes OADs in any region, and some other factors may accelerate the phenotype in the distal region. Because IFT machinery is known to disappear as the spermatozoa mature (San Agustin et al, 2015), we speculate that the supply of OAD during spermiogenesis decreases as the sperm flagella elongate, leading to the shortage of OAD in the distal flagella to compensate for the instability of *calaxin*$^{-/-}$ OADs. E130A-Calaxin rescued this phenotype (Fig 1C and D), indicating that Calaxin stabilizes OAD independent of calcium. Further supporting this idea, the expression of E130A-Calaxin rescued the abnormal motility of KV cilia and the laterality randomization of the *calaxin* $^{-/-}$ mutant (Fig 2A–D), which possibly resulted from the reduction of OAD. It is likely that the Calaxin knockout reduced OAD and the reduction resulted in incomplete bend propagation and abnormal movements of KV cilia and reactivated sperm flagella. The axoneme from human PCD patients also shows the reduction of OAD (Hjeij et al, 2023), suggesting such stabilizing function is conserved among vertebrates. It also implies that Calaxin does not work as a calcium-dependent dynein regulator in KV cilia. Consistently, calcium depletion does not alter KV ciliary motility (Yuan et al, 2015) and KV cilia exhibit uniform rotation without waveform switching.

How Calaxin-dependent dynein inhibition is transformed into an asymmetric waveform remains to be determined. One hypothesis is that the localization of OAD components is not uniform, resulting in the uneven inhibition of dynein activity. However, structural studies of sperm flagella did not show such asymmetry (Chen et al, 2023; Yamaguchi et al, 2023). If Calaxin localizes to specific DMTs, it is likely that loss of Calaxin does not affect overall OAD conformation. Instead, *calaxin* $^{-/-}$ sperm partly lacks OAD in their distal region, suggesting that Calaxin exists on all DMTs. Another possibility is that asymmetric axonemal structures such as o-SUB5-6 in sea urchin flagella (Lin et al, 2012) transform the global inhibition of dynein activity into an asymmetric waveform.

Another limitation of this study comes from the sperm model we used to precisely control the calcium condition. In WT, although the waveform of the sperm model was similar to that of intact sperm, it exhibited a much slower beating compared with intact sperm (Fig S2C). In *armc4*$^{-/-}$ sperm, the frequency of beating flagella was also strongly reduced (Fig S3C) (Yamaguchi et al, 2023). One possibility is that the reactivation solution does not completely mimic the intraciliary environment. Alternatively, the loss of plasma membrane may accelerate the disruption of axonemal dyneins under the depletion of Calaxin or Armc4. It suggests the demembranated sperm model is not a completely physiological system, and analyzing intact sperm obtained from mutants we used may contribute to further understanding.

# Materials and Methods

## Zebrafish maintenance

Zebrafish were maintained at 28.5°C. Embryos and larvae were raised at the same condition in 1/3 Ringer's solution. Embryos were staged as previously described (Kimmel et al, 1995).

## Plasmids

*calaxin* was amplified from *Danio rerio* testis cDNA with calaxin _cloning.F (GCAACATTCAACCAAAGACTAAAAAGAG) and calaxin_cloning.R (GAGGGAAAACAACAAATTGTCAAAC) primers and cloned into the pCR-Blunt vector. pCR-Blunt-calaxin was digested with EcoRI and subcloned into the pCS2+ EcoRI site. For protein expression, testis cDNA was amplified with calaxin_EcoRI.F (GGGAATTCCCATGCTGAAAATGTCGGC-GATG) and calaxin_EcoRI.R (GGGAATTCTTATTCTTTGCAGTGTTCGTGTTTCTG) primers, digested with EcoRI, and cloned into the pGEX 6P-2 (28-9546-50; Cytiva) EcoRI site. E130A mutation was introduced by PCR with calaxin_E130Amut.F (CACGAGAAGCCATGTTTCAGATGCTGAAAGACAGC) and calaxin_E130Amut.R (ACATGGCTTCTCGTGAGATGTATCCATCCCCG) primers. To generate a Tol2 donor plasmid, a 5.3-kb promoter element from the β-actin gene (Higashijima et al, 1997) was cloned from the zebrafish genome with actb2p5304_clo.F (AATTCCAGTTTGAAGAAACTTTTC) and actb2p5304_clo.R (GGCTGAACTGTAAAAGAAAGG) primers. The amplified fragment was combined with Tol2 elements and inserted into the pBlueScript SK(+) vector (Kawakami, 2007). The resulting pBlueScript SK(+)-Tol2-actb plasmid was amplified with pBSKTol2actb(calaxin).F and pBSKTol2actb(calaxin).R primers to add Kozak (GCCACC) sequence. Calaxin ORFs were amplified from pGEX 6P-2 calaxin WT or E130A with calaxin_ORF.F (ATGCTGAAAATGTCGGCGAT) and calaxin_ORF.R (TTATTCTTTGCAGTGTTCGTGTTTCT) primers. These fragments were ligated with NEBuilder (E2621; New England Biolabs). All plasmids were sequenced before they were used.

## mRNA synthesis

*calaxin* WT and E130A mRNA were transcribed from pCS2+-*calaxin* WT and E130A plasmids using the mMESSAGE mMACHINE SP6 kit (AM1340; Invitrogen) following the manufacturer's instruction. The template plasmid was digested with DNase I, and mRNA was

purified by ethanol precipitation. mRNA was quantified by Abs260 and stored at −80°C before use.

## Mutagenesis

To produce *Tg WT* or *Tg E130A*, *calaxin* $^{-/-}$ embryos were injected at the one-cell stage with 20 pg Tol2 transposase mRNA and 30 pg pBlueScript SK(+)-Tol2-actb-calaxin WT or E130A plasmid. Injected embryos were raised and crossed with *calaxin* $^{-/-}$ fish. F1 fish were screened for integration by PCR with calaxin_check.F (GGA-GAGCAGGCAGAGAGAAAG) and calaxin_D164_check.R (GGTTTTCATC-CATGACCGTCTTCTC) primers. PCR-positive male F1 fish were further subjected to sperm immunofluorescence with anti-Calaxin anti-body, and those with Calaxin-positive sperm were used for experiments.

## Sperm collection

To collect sperm, adult zebrafish were anesthetized with tricaine. Sperm were collected by pushing their abdomen or euthanizing and dissecting their testis in 5 µl PBS. From the pushing method, 0.1–0.5 µl of sperm was obtained from each fish.

## Immunofluorescence

The collected sperm were diluted with Hank's buffer so that 10–20 sperm/view were observed at x200 magnification. 40 µl of the sperm solution was placed on the glass slide coated with 0.01% polyethyleneimine for 10 min. After the solution was removed, the attached sperm were fixed and demembranated with Hank's buffer containing 4% PFA and 0.05% NP-40. The slide was washed and dehydrated with acetone followed by methanol at −20°C. Then, it was rehydrated with PBS, blocked with blocking buffer (PBS/0.05% Tween-20, 2% cold water fish gelatin, 4% normal goat serum), and incubated with primary antibodies (blocking buffer with 1/2,500 anti-acetylated alpha-tubulin antibody and 1/50 polyclonal anti-bodies) at 4°C overnight. After the wash with PBS/0.05% Tween-20 (PBST), it was incubated with secondary antibodies (blocking buffer with 2.5 µg/ml DAPI and 1/250 secondary antibodies corresponding to primary antibodies) at RT for 2 h and washed twice with PBST and once with PBS. Images were obtained with a fluorescence micro-scope (BX60; Olympus) equipped with a 40x NA0.75 objective lens and a CCD camera (ORCA-R2; Hamamatsu and Micro-Manager software).

## Zebrafish demembranated sperm model

Zebrafish sperm were suspended in 5 µl PBS. To demembranate the sperm, 1 µl of the suspension was added to 20 µl HMDK (30 mM Hepes−NaOH, pH 7.2, 5 mM MgSO$_4$, 1 mM DTT, 50 mM KAcO) with 0.1 mM EGTA and 0.5% NP-40. For activation and observation, 1 µl demembranated sperm and 10 µl of activation solution (HMDK, 0.5% NP-40, 0.1 mM EGTA or CaCl$_2$, 50 µM ATP, 50 µM ADP) were mixed for activation and recorded with a phase-contrast microscope (BX53; Olympus) equipped with a 100x NA1.35 objective lens and a high-speed camera (EoSens MC1362; Mikrotron and StreamPix 5 software) at 100 fps, 640 × 480 pixels. For inhibition of dynein with vanadate,

10 µM sodium orthovanadate was added to the activation solution. Demembranated sperm was kept on ice and used within 1 h after the sperm collection.

## Waveform analysis

Recorded videos of flagella were converted into 2,560 × 1920 and manually traced with ImageJ. Traces were analyzed with self-made Python programs (Fig S2E). First, the LineTrace_4x.py program calculated the tangent angle and the cumulative length from the head on every 10 pixels (~0.4 µm). To obtain the tangent angle, the tangent line was drawn as an average of 20 pixels (~0.8 µm). The tangent angle is the angle between the tangent line and the horizontal axis of the image. Next, tangent angles from $\geqq 5$ time points in one beating cycle were combined and fitted with the linear equation by the PlotLine.py program. The tangent angle plot and the fitted equation were shifted so that the equation starts from (x, y) = (0, 0). "Basal curvature" was defined as the slope of the fitted equation. To calculate the asymmetry index, sequential 20 pixels were fitted with a circle and the curvature was defined as 1/r, where r is the radius of the fitted circle. The asymmetry index was defined as the ratio of the maximal to the minimal value of the curvature at 10 µm from the head. We assumed the bend with the maximal curvature is "P-bend" and the one with the minimal curvature is "R-bend." Python scripts we used are available on GitHub (https://github.com/motohiromorikawa/Calaxin-analysis).

## KV ciliary motility

Zebrafish Kupffer's vesicle was observed as previously described (Sasaki et al, 2019). Briefly, 8–10 somite embryo was fixed on a glass dish by 1.0% low melting point agar in 1/3 Ringer's solution, recorded with a bright-field microscope (DMI6000B; Leica) equipped with a 100x NA1.4 objective lens and a high-speed camera (HAS-L1; DITECT and HAS-L1 Basic software) at 1,000 fps. For mRNA-injected embryos, 10 pg of mRNA was injected at the one-cell stage.

## Purification and ITC experiment of Calaxin

*E. coli* BL21 cells transformed with pGEX 6P-2 *calaxin* WT or E130A were cultured in LB medium at 37°C to reach Abs 0.5 at 600 nm, followed by induction with 0.1 mM IPTG. For WT, cells were cultured at 37°C for 3 h after the induction. For E130A, they were cultured at 20°C overnight. Harvested cell pellets were sonicated, clarified, and glutathione-affinity–purified (GST-Accept, 09277; Nacalai Tesque) in lysis buffer (PBS with 150 mM NaCl and 1 mM DTT) supplemented with 0.2x protease inhibitor cocktail (03969; Nacalai Tesque). The column was sequentially washed with lysis buffer with 0.01% NP-40 and 0.2x protease inhibitor, lysis buffer with 0.01% NP-40, 1 mM EDTA, and 0.2x protease inhibitor, lysis buffer with 0.01% NP-40 and 1 mM EDTA, lysis buffer with 0.005% NP-40, and lysis buffer with 0.001% NP-40. After the wash, lysis buffer with 0.001% NP-40 and HRV 3C Protease (7360; TaKaRa) was added for on-column digestion at 4°C overnight. The eluted protein was applied to TALON Metal Affinity Resin (635501; TaKaRa) to remove the protease. The resulting protein was concentrated, and the buffer was exchanged for storage buffer (PBS with 150 mM NaCl) by Vivaspin 2–10K (VS0201;

Sartorius), then stored at −80°C. To check the quality and mobility of recombinant proteins, SDS–PAGE was conducted with Extra PAGE One Precast Gel (13064-64; Nacalai Tesque). Proteins were mixed with the sample buffer containing 1 mM $CaCl_2$ or EGTA and denatured at 95°C for 5 min before electrophoresis. Before the ITC experiment, Calaxin was supplemented with 0.5 mM EGTA to remove residual calcium and buffer-exchanged on Superdex 200 10/300 GL (17517510; Cytiva) equilibrated with ITC buffer (25 mM MOPS-KOH, pH 7.8, 200 mM NaCl, 1 mM DTT). MicroCal iTC200 and Origin software (GE Life Sciences) were used for the ITC experiment and analysis as previously described (Mizuno et al, 2009).

### Antibodies

To visualize the axoneme, a mouse monoclonal anti-acetylated alpha-tubulin antibody (T6793; Sigma-Aldrich) was used. Other polyclonal primary antibodies are as follows: guinea pig anti-Calaxin antibody (Sasaki et al, 2019) (referred to as anti-Efcab1 antibody), rabbit anti-Dnah2 antibody (Yamaguchi et al, 2018), rabbit anti-Dnah8 antibody (Yamaguchi et al, 2018), and rabbit anti-Dnah9 antibody (Yamaguchi et al, 2023). As secondary antibodies, goat anti-rabbit IgG Alexa Fluor 488 (A-11008; Invitrogen), goat anti-guinea pig IgG Alexa Fluor 488 (A-11073; Invitrogen), and goat anti-mouse IgG Alexa Fluor 555 (A-21422; Invitrogen) were used.

### Statistics

Statistical analyses were conducted with R version 4.1.2. Statistical significances were tested by an unpaired two-tailed $t$ test. For multiple comparisons, $P$-values were adjusted by the Holm–Bonferroni method. $P$-value < 0.05 was considered to be significant.

## Data Availability

The data are available from the corresponding author upon reasonable request.

## Supplementary Information

## Acknowledgements

We would like to thank Osamu Nureki, Atsuhiro Tomita, and Sonomi Yamaguchi for their assistance in the ITC experiment and data analysis. This research was supported by the Japan Society for the Promotion of Science KAKENHI grant numbers 16H02502, 21H04762, and 21H05248 to M Kikkawa.

## Author Contributions

M Morikawa: conceptualization, software, formal analysis, investigation, visualization, and writing—original draft, review, and editing.

H Yamaguchi: formal analysis, investigation, and writing—original draft, review, and editing.

M Kikkawa: conceptualization, supervision, funding acquisition, project administration, and writing—review and editing.

### Conflict of Interest Statement

The authors declare that they have no conflict of interest.

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
