## [Reviewer comments · Life Science Alliance]

Life Science Alliance

Calaxin is a key factor for calcium-dependent waveform control in zebrafish sperm

Motohiro Morikawa, Hiroshi Yamaguchi, and Masahide Kikkawa

DOI: <https://doi.org/10.26508/lsa.202402632>

Corresponding author(s): Masahide Kikkawa, The University of Tokyo

Review Timeline:

Submission Date:	2024-01-30
Editorial Decision:	2024-03-18
Revision Received:	2024-05-24
Editorial Decision:	2024-05-28
Revision Received:	2024-05-30
Accepted:	2024-06-04

Transaction Report:

March 18, 2024

Re: Life Science Alliance manuscript #LSA-2024-02632-T

Prof. Masahide Kikkawa
The University of Tokyo
Department of Structural Biology, Graduate School of Medicine
7-3-1 Hongo
Bunkyo, Tokyo 113-0033
Japan

Dear Dr. Kikkawa,

Thank you for submitting your manuscript entitled "Calaxin is a key factor for calcium-dependent waveform control in zebrafish sperm" to Life Science Alliance. The manuscript was assessed by expert reviewers, whose comments are appended to this letter. We invite you to submit a revised manuscript addressing the Reviewer comments.

Thank you for this interesting contribution to Life Science Alliance. We are looking forward to receiving your revised manuscript.

Sincerely,

B. MANUSCRIPT ORGANIZATION AND FORMATTING:

Reviewer #1 (Comments to the Authors (Required)):

The authors demonstrate two distinct functions of calaxin using zebrafish: Ca²⁺-dependent regulation of flagellar asymmetry in sperm and Ca²⁺-independent stabilization of KV cilia. The experiments are all well designed and performed and thus the paper greatly contributes in understanding the function of calaxin in ciliary motility. I would simply like to recommend the authors to address following issues before publication.

1. It is difficult to determine the sperm head axis in teleost sperm with round heads. The authors thus employed the tangent angle in zebrafish sperm in flagellar analysis. This is reasonable and the results well describe the difference in wave propagation and asymmetry between WT and calaxin ^{-/-} sperm. The abnormal wave propagation at the tip region is also obvious in calaxin ^{-/-} sperm. The authors showed the sperm flagellar asymmetry by basal curvature. This way may indirectly describe the asymmetric waves but does not directly express the asymmetric properties between the principal and reverse bend. The authors may add the "asymmetry index", which is the ratio of maximal curvatures of both bends (Mizuno et al, 2012).
2. In zebrafish the demembrated sperm are not fully reactivated under high concentration ATP (>0.2mM). However, many studies show that OAD functions in increasing the beat frequency of a flagellum under high concentration of ATP. Data under high concentration ATP should strengthen this paper.
3. It is interesting that the calaxin KO affects OAD stability, notably observed in the tip or distal region of sperm flagella (Figure 1, immunofluorescence). Why is OADs in the proximal most region normally assembled without calaxin and only a limited distal region affected?
4. The kymographs in Fig.2C show the independency of Ca²⁺ binding to calaxin in the motility of KV cilia and the zebrafish body laterality. Beat frequency of calaxin ^{-/-} flagella shows lower than those of WT and E130A mRNA injected embryos. However, the irregularly vibrating movement is observed in calaxin ^{-/-} zebrafish and also in calaxin ^{-/-} sperm. Is this due to the lack of calaxin function in bend propagation?
5. Figure 3A: The caption for the colors could not been linked to the immunofluorescence, instead to the tangent angle plot. The authors had better to layout the position of caption and immunofluorescent image.
6. Line 177: 50 μ M ATP 50 μ M ADP condition; may be better to insert "plus" or "-" between ATP and ADP? This also appears in other places in the text and figure.
7. Line 200: "relative to the recorded frame" is unclear. Is it the tangent angle along a flagellum in each frame recorded?

Reviewer #2 (Comments to the Authors (Required)):

This manuscript describes an experimental study of the role of Calaxin in stabilizing outer dynein arms and influencing asymmetry of the flagellar waveform of Zebrafish sperm. Using a transgenic mutant in which Calaxin is deficient in calcium-binding, outer dynein arms are stabilized but calcium-induced waveform asymmetry is not recapitulated, the authors show that Calaxin is a calcium-dependent regulator of dynein but calcium independent stabilizer of outer arm dynein. The studies are well designed and the data support the conclusions.

My request to the authors is to clarify and provide more details on the waveform analysis described briefly in lines 198-202. "Instead, we selected sperm with their heads fixed to the glass slide and freely swimming flagella and calculated the tangent angle relative to the recorded frame. To obtain the time- and spatial average, we plotted tangent angles from ≥ 5 traces over one beat cycle followed by the fitting with the linear equation. Basal curvature was defined as the slope of the fitted equation."

Please

- Clarify what they mean by "the tangent angle relative to the recorded frame."

- Describe the tools or algorithm used to trace the waveform and calculate the tangent angle. Is it automated? Publicly available? Were there any smoothing involved?
- Provide more details on the waveform analysis in Methods or Supplementary Material

Dear Dr. Eric Sawyer, Executive Editor, Life Science Alliance

It is our pleasure to submit the revised version of our manuscript: “**Calaxin is a key factor for calcium-dependent waveform control in zebrafish sperm**”, to Life Science Alliance. Compared to the original submitted version, we have improved the descriptions and analyses, thanks to the comments and suggestions by reviewers.

Specific comments and answers are summarized below. Comments and suggestions by reviewers were valuable in clarifying key points of our manuscripts. We believe that the current version of the manuscript will meet the requirements of Life Science Alliance.

Sincerely yours,

Masahide Kikkawa, Ph.D.

Professor

The University of Tokyo

7-3-1 Hongo, Bunkyo, Tokyo, Japan

Tel. & Fax: +81-3-5841-3339

E-mail: mkikkawa@m.u-tokyo.ac.jp

Answers to Reviewer #1's comments

Reviewer #1 (Comments to the Authors (Required)):

1. It is difficult to determine the sperm head axis in teleost sperm with round heads. The authors thus employed the tangent angle in zebrafish sperm in flagellar analysis. This is reasonable and the results well describe the difference in wave propagation and asymmetry between WT and calaxin -/- sperm. The abnormal wave propagation at the tip region is also obvious in calaxin -/- sperm. The authors showed the sperm flagellar asymmetry by basal curvature. This way may indirectly describe the asymmetric waves but does not directly express the asymmetric properties between the principal and reverse bend. The authors may add the "asymmetry index", which is the ratio of maximal curvatures of both bends (Mizuno et al, 2012).

To answer this comment, we calculated “asymmetry index” as in Mizuno et al., 2012 and created Supplementary Figure 2D. We added lines 215-222 to describe the result.

“We also calculated the asymmetry index as in Ciona sperm (Mizuno et al., 2012) (Fig S2D). The result showed a similar trend as the basal curvature: the pCa4 condition led to higher asymmetry indices, with *Tg EI30A* exhibiting a weaker elevation compared to WT and *Tg WT*. However, these differences did not reach statistical significance, possibly due to the limited sample size. Note that we utilized an alternative definition of 'P-bend' and 'R-bend': we defined the beating with higher curvature as 'P-bend' and the other as 'R-bend', deviating from the original definition based on the sperm swimming path, as we observed sperm fixed to the glass slide.”

2. In zebrafish the demembrated sperm are not fully reactivated under high concentration ATP (>0.2mM). However, many studies show that OAD functions in increasing the beat frequency of a flagellum under high concentration of ATP. Data under high concentration ATP should strengthen this paper.

Thank you for the suggestion. We agree that higher ATP concentration is more desirable to fully reproduce the sperm beating frequency in the reactivated sperm model. However, in zebrafish, we found that higher ATP concentration reduced the reactivation rate, especially in the presence of calcium. Supplementary Figure 2B illustrates the reactivation rate under different ATP concentrations. In the pCa4 condition, 100 μ M or higher ATP + 50 μ M ADP reactivated less than half of the sperm, while almost 2/3 of them were reactivated in 50 μ M ATP + 50 μ M ADP. In the EGTA condition, the reactivation rate was better, but higher ATP concentration gradually decreased it. Lower reactivation rate a) made the data collection more difficult, because only a small portion of reactivated sperm had freely swimming tails (demembrated flagella easily adhered to the glass slide), and b) raised the possibility that the sperm flagella is not properly reactivated. Thus, we decided to use 50 μ M ATP + 50 μ M ADP condition, where more than half of sperm were reactivated under the presence of calcium or EGTA.

To clarify these points, we modified lines 175-181 as follows.

(before)

Comparison of different ATP and ADP concentrations showed that higher ATP concentration induced faster beating frequency and that the supplementation of ADP reduced the variation of beating frequencies. We chose 50 μ M ATP 50 μ M ADP condition that induced relatively slow, uniform beating, which made observation easier (Fig. S2 A).

(after)

Comparison of different ATP and ADP concentrations showed that higher ATP concentration induced faster beating frequency and that the supplementation of ADP reduced the variation of beating frequencies (Fig S2A). On the other hand, higher ATP concentration resulted in lower reactivation rate especially in the pCa4 condition (Fig S2B). Thus, we chose 50 μ M + ATP 50 μ M ADP condition to enable the stable beating frequency and the higher reactivation rate.

3. It is interesting that the calaxin KO affects OAD stability, notably observed in the tip or distal region of sperm flagella (Figure 1, immunofluorescence). Why is OADs in the proximal most region normally assembled without calaxin and only a limited distal region affected?

In our recent paper (Yamaguchi et al., 2023), we analyzed the detailed distribution of *calaxin*^{-/-} OADs using cryo-electron tomography. We revealed that the calaxin KO induces OAD-missing clusters at various regions of the flagella (i.e., not only the distal region but also the proximal region), with a gradual increase and expansion of the clusters towards the distal end. This suggests that Calaxin KO destabilizes OADs at any region of flagella, but some other factors

may accelerate the phenotype in the distal region. We propose a hypothesis that the supply of OAD during spermiogenesis decreases as the sperm flagella elongate, leading to the shortage of OAD in the distal flagella to compensate for the instability of calaxin^{-/-} OADs. This hypothesis is supported by the known disappearance of IFT machinery as the spermatozoa mature (San Agustin et al., 2015).

To discuss this point, we added the following sentences in lines 285-291:

“Intriguingly, we have previously shown that OADs are also reduced in the proximal region of calaxin^{-/-} flagella (Yamaguchi et al., 2023). This indicates that Calaxin knockout destabilizes OADs in any region, and some other factors may accelerate the phenotype in the distal region. Since IFT machinery is known to disappear as the spermatozoa mature (San Agustin et al., 2015), we speculate that the supply of OAD during spermiogenesis decreases as the sperm flagella elongate, leading to the shortage of OAD in the distal flagella to compensate for the instability of calaxin^{-/-} OADs.”

4. The kymographs in Fig.2C show the independency of Ca²⁺ binding to calaxin in the motility of KV cilia and the zebrafish body laterality. Beat frequency of calaxin^{-/-} flagella shows lower than those of WT and E130A mRNA injected embryos. However, the irregularly vibrating movement is observed in calaxin^{-/-} zebrafish and also in calaxin^{-/-} sperm. Is this due to the lack of calaxin function in bend propagation?

We speculate that the irregular movements of calaxin^{-/-} KV cilia and sperm flagella came from

the loss of Calaxin, which reduced OAD and eventually resulted in incomplete bend propagation.

To clarify this point, we added the following sentence in lines 295-297:

“It is likely that the Calaxin knockout reduced OAD and the reduction resulted in incomplete bend propagation and abnormal movements of KV cilia and reactivated sperm flagella.”

5. Figure 3A: The caption for the colors could not been linked to the immunofluorescence, instead to the tangent angle plot. The authors had better to layout the position of caption and immunofluorescent image.

Thank you for the comment. We moved the caption below the immunofluorescent image.

6. Line 177: 50 μ M ATP 50 μ M ADP condition; may be better to insert "plus" or "-" between ATP and ADP? This also appears in other places in the text and figure.

We changed the wording to “50 μ M ATP + 50 μ M ADP condition” in lines 173, 179 and Supplementary Figure 3.

7. Line 200: "relative to the recorded frame" is unclear. Is it the tangent angle along a flagellum in each frame recorded?

We modified Supplementary Figure 2E and added the "Waveform analysis" paragraph in the Materials and Methods section. The middle panel of Supplementary Figure 2E describes how we defined the tangent angle. It is the angle between the tangent line and the horizontal axis of the recorded movie frame.

Answers to Reviewer #2's comments

Reviewer #2 (Comments to the Authors (Required)):

This manuscript describes an experimental study of the role of Calaxin in stabilizing outer dynein arms and influencing asymmetry of the flagellar waveform of Zebrafish sperm. Using a transgenic mutant with in which Calaxin is deficient in calcium-binding, outer dynein arms are stabilized but calcium-induced waveform asymmetry is not recapitulated, the authors show that Calaxin is a calcium-dependent regulator of dynein but calcium independent stabilizer of outer arm dynein.

The studies are well designed and the data support the conclusions.

My request to the authors is to clarify and provide more details on the waveform analysis described briefly in lines 198-202.

"Instead, we selected sperm with their heads fixed to the glass slide and freely swimming flagella and calculated the tangent angle relative to the recorded frame. To obtain the time- and spatial average, we plotted tangent angles from ≥ 5 traces over one beat cycle followed by the fitting with the linear equation. Basal curvature was defined as the slope of the fitted equation."

Thank you for the comment. As described in response to Reviewer #1's comments, we modified Supplementary Figure 2E and added "Waveform analysis" paragraph in the Materials and Methods section to describe the analysis in detail.

Please

- Clarify what they mean by "the tangent angle relative to the recorded frame."

To measure the tangent angle, we first drew a tangent line at each point. Next, we calculated the relative angle between the tangent line and the horizontal axis of the recorded movie frame. The middle panel of Supplementary Figure 2E shows these processes.

- Describe the tools or algorithm used to trace the waveform and calculate the tangent angle. Is it automated? Publicly available? Were there any smoothing involved?

As a first step, we manually traced the flagellum (not by an automated program) and drew tangent lines with a self-made Python script. Tangent lines were calculated as an average of ~0.8 um region. No other smoothing method was applied.

- Provide more details on the waveform analysis in Methods or Supplementary Material

We modified Supplementary Figure 2E and added “Waveform analysis” paragraph in the Materials and Methods section. Python programs we used are now available on GitHub (<https://github.com/motohiromorikawa/Calaxin-analysis>).

May 28, 2024

RE: Life Science Alliance Manuscript #LSA-2024-02632-TR

Prof. Masahide Kikkawa
The University of Tokyo
Department of Structural Biology, Graduate School of Medicine
7-3-1 Hongo
Bunkyo, Tokyo 113-0033
Japan

Dear Dr. Kikkawa,

Thank you for submitting your revised manuscript entitled "Calaxin is a key factor for calcium-dependent waveform control in zebrafish sperm". We would be happy to publish your paper in Life Science Alliance pending final revisions necessary to meet our formatting guidelines.

- please be sure that the authorship listing and order is correct
- please add ORCID ID for the corresponding -- you should have received instructions on how to do so
- please add the Twitter handle of your host institute/organization as well as your own or/and one of the authors in our system
- please be sure to mention all co-authors in the authors' contribution section and that their contributions match with those listed in the system

A. FINAL FILES:

B. MANUSCRIPT ORGANIZATION AND FORMATTING:

Sincerely,

June 4, 2024

RE: Life Science Alliance Manuscript #LSA-2024-02632-TRR

Prof. Masahide Kikkawa
The University of Tokyo
Department of Structural Biology, Graduate School of Medicine
7-3-1 Hongo
Bunkyo, Tokyo 113-0033
Japan

Dear Dr. Kikkawa,

Thank you for submitting your Research Article entitled "Calaxin is a key factor for calcium-dependent waveform control in zebrafish sperm". It is a pleasure to let you know that your manuscript is now accepted for publication in Life Science Alliance. Congratulations on this interesting work.

DISTRIBUTION OF MATERIALS:

Again, congratulations on a very nice paper. I hope you found the review process to be constructive and are pleased with how the manuscript was handled editorially. We look forward to future exciting submissions from your lab.

Sincerely,
